# Hair-Based Assessment of Sex Steroid Hormones in Patients with Anorexia Nervosa

**DOI:** 10.3390/metabo13010021

**Published:** 2022-12-22

**Authors:** Victoria-Luise Batury, Friederike I. Tam, Inger Hellerhoff, Marie-Louis Wronski, Katrin Borucki, Kerstin Weidner, Veit Roessner, Wei Gao, Stefan Ehrlich

**Affiliations:** 1Division of Psychological and Social Medicine and Developmental Neurosciences, Translational Developmental Neuroscience Section, University Hospital C.G. Carus, Technische Universität Dresden, 01307 Dresden, Germany; 2Eating Disorder Treatment and Research Center, Department of Child and Adolescent Psychiatry, Faculty of Medicine, University Hospital C.G. Carus, Technische Universität Dresden, 01307 Dresden, Germany; 3Institute of Clinical Chemistry and Pathobiochemistry, Otto-von-Guericke-University Magdeburg, 39120 Magdeburg, Germany; 4Department of Psychotherapy and Psychosomatic Medicine, Faculty of Medicine, University Hospital C.G. Carus, Technische Universität Dresden, 01307 Dresden, Germany; 5Department of Child and Adolescent Psychiatry and Psychotherapy, University Hospital C.G. Carus, Technische Universität Dresden, 01307 Dresden, Germany; 6Department of Psychology, Technische Universität Dresden, 01062 Dresden, Germany

**Keywords:** anorexia nervosa, hair sample analysis, sex steroid hormones, progesterone, DHEA

## Abstract

Anorexia nervosa (AN) is a complex psychiatric disorder accompanied by a variety of endocrine effects. Altered levels of the sex steroid hormones progesterone and dehydroepiandrosterone (DHEA) have been shown to occur in patients with AN using short-term hormonal measurement methods based on blood, saliva, and urine samples. However, since sex steroid hormone levels fluctuate during the menstrual cycle, these measurement methods require a great deal of effort due to the need to collect multiple samples in order to correctly determine the basal level of sex hormones. In contrast, hair-based assessments provide a marker of accumulated longer-term hormone exposure using a single, non-invasive sample. The aim of this study was to investigate sex steroid hormone levels via hair-based assessments in acutely underweight AN in comparison with healthy, age-matched, female control participants. Additionally, we compared progesterone and DHEA hair levels longitudinally during inpatient treatment in AN. Collected hair samples were analyzed using liquid chromatography-mass spectrometry (LC-MS/MS) to determine a monthly hormone level of progesterone and DHEA. Our results indicate that DHEA hair hormone levels were similar across groups but progesterone was suppressed in underweight AN compared with healthy controls. In the longitudinal design, no significant change in hair hormone levels during partial weight restoration in patients with AN was observed. Our findings suggest that hair analysis can be used to detect suppressed progesterone levels in severe AN, and that progesterone does not increase during short-term weight restoration.

## 1. Introduction

Anorexia nervosa (AN) is a life-threatening illness characterized by an intense fear of weight gain, body image distortion, and a reduction of food intake that results in patients becoming significant underweight. The disorder is accompanied by a multitude of endocrine dysregulations and metabolic changes as well as somatic complications [1,2]. One of the best-known endocrine manifestations of acute undernutrition in AN is hypothalamic suppression, which leads to dysfunction of the hypothalamic–pituitary–adrenal (HPA) and –gonadal (HPG) axes [3,4]. Moreover, this dysfunction also leads to impaired sex steroidogenesis (biosynthesis of sex steroid hormones) and may cause amenorrhea, among other symptoms [5].

Progesterone is one of the sex steroid hormones predominantly produced by the adrenal glands and gonads. It is known as the main female reproductive hormone, and specifically, it regulates uterine receptivity, implantation, and maintenance of pregnancy. Based on short-term hormone measurements, women with AN are known to have decreased circulating progesterone concentrations [6]. Furthermore, functional hypothalamic amenorrhea is characterized by a chronic suppression of this hormone [7].

Dehydroepiandrosterone (DHEA) is best known for its role as a sex steroid hormone precursor of androgens and estrogens. However, DHEA itself also exerts weak androgenic effects and acts as an important neurosteroid in the central nervous system. Moreover, DHEA has metabolic, immunomodulatory, endocrine, and musculoskeletal effects [8]. The majority of DHEA is produced in the adrenal glands, with the remaining synthesis occurring in the gonads and a small amount in the brain. However, existing data regarding DHEA concentrations in AN are inconsistent. Previous research has reported decreased [9,10], elevated [11,12] as well as normal DHEA concentrations in blood samples [13,14]. However, a recently published meta-analysis of selected studies concluded that elevated blood levels of DHEA are present in AN [15]. The heterogeneity of the results of previous studies on DHEA may be partially explained by their relatively small sample sizes, risks of confounding bias (i.e., alcohol consumption, smoking, BMI) as well as the applied measurement methods based on blood, saliva, and urine samples.

Short-term samples of steroid hormones capture moment-to-moment fluctuations and are easily influenced by the daily circadian pattern of adrenocorticotropic hormone (ACTH) secretion [16], emotional states, and stress. Progesterone in particular fluctuates widely during the ovulatory cycle. Therefore, short-term measurement methods require multiple samples to correctly estimate basal levels of sex steroid hormones [17].

An important biological indicator of recovery during the treatment of AN is the resumption of menses [18]. Weight restoration is an important prerequisite for reestablishing endocrine functions, especially in the HPG axis. Studies examining underlying factors and predictors of menstrual recovery in AN have mostly employed follow-up assessments many months or even years after the acute AN episode [19,20,21,22]. However, the effects of short-term weight restoration on the HPG axis and associated hormones, in particular progesterone and DHEA concentrations, are yet poorly understood. A number of endocrine mediators, such as leptin and ghrelin concentrations, that are implicated in this complex process may modulate or interact with gonadotropin secretion (i.e., luteinizing hormone (LH) and follicle-stimulating hormone (FSH), which stimulate the gonads) and should, therefore, be taken into account [23].

The analysis of hormone levels in hair is a method that uses non-invasive sampling to provide a marker of the accumulated longer-term (1–3 months) hormone exposure. In contrast to more conventional salivary and plasma measures, hair steroid profiling is not confounded by short-term temporal variation (circadian, menstrual cycle) or hormonal reactivity [24,25]. By quantifying the integrated hormone concentration accumulated over one up to several months, longer-term hormone exposure can be assayed before, over and after the time of treatment in AN. Numerous studies have used hair sample-based sex steroid hormones successfully to examine expected longer-term changes in specific cohorts, e.g., patients with depression and anxiety disorders as well as normal-weight individuals during puberty [24,25]. In contrast to the repeated collection of blood or saliva samples, the measurement of sex steroid hormones in hair is more compatible with and much more practical for everyday clinical use, especially in underweight patients.

In this study, we investigated hair sample-based data on the sex steroid hormones progesterone and DHEA. The initial aim was to examine whether the altered sex steroid hormone levels, shown in acutely underweight AN using conventional methods, can also be shown by means of hair analyses. Therefore, we compared longer-term hair progesterone and hair DHEA in acutely ill patients with AN and healthy control participants (HC). To gain a broader understanding of the impact of treatment-induced weight gain on hormone concentrations, we also measured hair hormone concentrations of patients with AN before and after approximately three months of inpatient treatment in a longitudinal design. Given the established role of other hormones interacting with the HPA and HPG axes in AN, we further tested the hypothesis that leptin and ghrelin concentrations would correlate with hair sample-based measured progesterone and DHEA concentrations.

## 2. Materials and Methods

### 2.1. Study Participants

The sample population consisted of two groups: 33 female patients with AN in the acutely underweight state (12–28 years old) and 33 female HC (12–27 years old), matched to AN by age using an optimal pair matching algorithm [26]. Patients with AN were admitted to intensive treatment within an eating disorder program at a child and adolescent psychiatry and psychosomatic medicine department of a tertiary care university hospital. AN was diagnosed with the expert form of the Structured Interview for Anorexia and Bulimia Nervosa (SIAB-EX [27]), adapted to meet DSM-5 criteria and supplemented with medical records and our own semi-structured interview, and required a BMI < 17.5 kg/m^2^ (or < 10th age percentile if younger than 18 years). HC had to be of normal weight, eumenorrheic, and without any history of psychiatric illness. HC were recruited through advertisements among middle school, high school, and university students. While HC were only assessed once, patients with AN were assessed within 96 h of admission to intensive treatment (AN-T1) and reassessed after short-term weight restoration (AN-T2, *n* = 29 of 33, BMI increase of at least 12%) for the longitudinal arm of this study. Reasons for loss to follow-up included insufficient weight gain, premature discharge, psychopharmacological treatment, and withdrawal of consent. Information regarding exclusion criteria and possible confounding variables, including menstrual cycle and use of contraceptive medication, were obtained from all participants using the SIAB-EX [27], supplemented by our own semi-structured interview and medical records. Comorbid diagnoses were derived according to standard practice from medical records and confirmed by an expert clinician with over 10 years of experience. Because the aim of the study was to examine endogenous variation in hormones, participants taking hormonal contraception were not included (for further exclusion criteria see Appendix A). All protocols received ethical approval from the local Institutional Review Board and all participants (or their guardians, if under 18 years old) gave written informed consent after full explanation of the purpose and nature of all procedures used.

### 2.2. Clinical Measures

In addition to the evaluation with the SIAB-EX [27], eating disorder-specific psychopathology was assessed with the German version of the self-report questionnaire Eating Disorder Inventory-2 (EDI-2) [28]. Depressive symptoms were explored using the German version of the Beck Depression Inventory-II (BDI-II) [29] and general levels of psychopathology and anxiety symptoms using the revised Symptom Checklist 90 (SCL-90-R) [30]. In addition to BMI, the age-/sex-corrected BMI standard deviation score (BMI-SDS) was calculated [31,32]. Demographic and clinical study data were collected and managed using the secure, web-based electronic data capture tool REDCap (Research Electronic Data Capture) [33].

### 2.3. Hair Sampling Procedure and Biochemical Analysis

Participants were instructed not to use any hair products that were not rinsed out of the hair on the day of the appointment. Hair samples were taken from the posterior vortex region with the hair cut as close as possible to the skin as recommended by Wennig [34]. The first 3 cm of hair (weight = 7.5 ± 0.5 mg) were analyzed as a marker of hormone accumulation over the last 3-month period based on an average hair growth rate of 1 cm/month [34] (hair sample 1, cross-sectional design). After 14 weeks, a second hair sample was taken in patients with AN as described above (hair sample T2, longitudinal design). The 3 cm hair segment taken after 14 weeks (T2_1–3_) was divided into three 1 cm segments to identify average hormone concentration per month. The hair follicle already grew beneath the scalp (length of 3–4 mm; [35]); therefore, we included an additional latency time of 2 weeks (12 weeks/3 months + 2 weeks), assuming that the hair needs 2 weeks to grow above the scalp (see Figure 1, latency period). Subsequently, hair progesterone and hair DHEA levels were assessed for each patient at 4 different time points. T1, representing the pre-treatment hormone levels, was calculated as the average of the 3-month period before treatment. Hair samples T2_1_, T2_2_, and T2_3_ represented the respective average hormone levels of each month of therapy. The hair samples were stored in aluminum foil packets at room temperature until analyzed with liquid chromatography-tandem mass spectrometry (LC-MS/MS [36]), a highly sensitive, selective, and reliable procedure for the quantification of hormone concentrations in hair samples. In line with the analytical protocol of Gao et al. [36], hair progesterone extraction procedures were conducted as follows. Samples were washed in 2.5 mL isopropanol for 3 min and dried for 12 h. After that, 7.5 ± 0.5 mg of non-pulverized hair per sample was weighed out. The samples were incubated in 1.8 mL methanol at room temperature for 18 h, and then 1.6 mL of clear supernatant was transferred into a glass vial. Subsequently, methanol was evaporated at 45 °C under a steady stream of nitrogen. The internal standard mixture was prepared in methanol at the final concentrations (progesterone-d9: 1.8 ng/mL, DHEA-d4: 2.1 ng/mL). For LC-MS analysis, 200 μL of the standard mixture was used.

### 2.4. Plasma Hormone Measures

For leptin and ghrelin assessment, venous blood samples were collected into vacutainer tubes between 7 and 9 a.m. after an overnight fast. In the AN group, venipuncture at T1 took place within 96 h after initiating intensive treatment. Aprotinin was added directly after blood sampling to prevent protein degradation by serine proteases. The blood samples were then immediately centrifuged (2500× *g* for 15 min) in a pre-cooled centrifuge (5 °C), aliquoted and stored at −80 °C. The plasma leptin and deacylated ghrelin concentrations were measured using commercially available enzyme-linked immunosorbent assay (ELISA, BioVendor Research and Diagnostic Products, Brno, Czech Republic).

### 2.5. Statistical Analyses

To address possible age-related effects on pubertal hormones in the cross-sectional study, we applied an optimal pairwise age-matching for AN and HC with a maximum age distance of 2 years within matched AN-HC pairs [26]. Data were initially collected from 87 participants (42 AN and a total of 45 HC), resulting in a sample of 66 female volunteers after pairwise age-matching: 33 AN (12.1–28.5 years) and 33 female age-matched HC (12.1–27.6 years). For age-matching, statistical analyses were conducted using R Software version 4.0.3 with the “MatchIt” package. All other statistical analyses were performed using IBM SPSS Statistics for Windows, version 27 (IBM Corp., Armonk, NY, USA).

Due to deviations from normality of the measured hair hormones and the high number of values below the limit of detection, we used non-parametric statistics for all hair hormone analyses. The lowest measurable value (defined as the limit of detection, LOD) was 0.1 pg/mg for progesterone and DHEA. Based on the assumption that values below the lower limit of detection indicate true values [37], all data points with below-threshold values were included in all models. These values were replaced by the LOD divided by the square root of 2, which is a common single-value imputation method [38]. In the cross-sectional analysis, mean hair progesterone and hair DHEA was compared between AN-T1 and HC with the Mann–Whitney U Test. For our longitudinal analysis, we conducted a Friedman test of differences among repeated measures to investigate possible hormone changes in the AN group across 4 time periods ranging from before admission to the end of intensive treatment (see Figure 1). To verify our results, we used a Chi-square test to examine the association between data points below/above LOD and groups AN-T1/HC (cross-sectional) or between data points below/above LOD and 4 time periods (longitudinal), respectively. Finally, partial Spearman correlations, adjusted for age, were calculated to investigate possible associations between hair hormone levels and leptin and ghrelin concentrations and the clinical variables BMI-SDS, BDI-II total score, EDI-2 total score, and SCL-90-R global severity index. Correlations were only calculated for sex steroid hormones with statistically significant group differences in previous analyses. Statistical significance was defined as *p* < 0.05. *p* values were adjusted for multiple comparisons across 6 tests using the false discovery rate (FDR) correction method [39].

## 3. Results

The demographic and clinical characteristics are summarized in Table 1 for the cross-sectional sample (AN-T1, HC) and in Table 2 for the longitudinal sample (AN-T1, AN-T2_1–3_). As both groups in the cross-sectional sample were closely matched, no difference regarding age was observed. As expected, AN-T1 had significantly lower BMI-SDS, higher levels of psychopathology (EDI-2, BDI-II, SCL-90-R) as well as higher ghrelin and lower leptin plasma concentrations than HC, which reflects the acutely undernourished state of AN. In the longitudinal sample, AN gained 0.93 kg per week on average, resulting in an average weight gain of 11.1 kg (26.4%) over the time of the study. Over the course of short-term weight restoration, a significant increase in leptin plasma concentrations and a decrease in ghrelin were observed (Table 2).

For the hair concentrations of progesterone, but not DHEA, the Mann–Whitney U Test indicated significantly lower levels in AN-T1 compared with HC (Table 2). Here, a more basic, categorical examination of values below vs. above LOD showed that hair progesterone in AN-T1 was significantly more often below LOD (54.5%) compared with HC (21.2%). For hair DHEA, the difference was not significant (for details see Appendix A).

In our longitudinal AN sample, there was no significant difference for the progesterone and DHEA hair concentrations across the four time points (T1, T2_1–3_; Table 2; for details, see Appendix A). A categorical examination showed that 58.62–65.52% of the values for hair progesterone and 10.34–13.79% of the values for hair DHEA were below LOD, which did not differ across time periods (for details, see Appendix A).

At the end of treatment, we observed a still significantly lower hair progesterone concentration in AN-T2_3_ compared with HC but no differences in hair DHEA (see Appendix A).

Partial correlation analyses between hair progesterone level and clinical variables across AN-T1 and HC (combined sample) revealed associations with leptin (r(54) = 0.422, *p _adjusted_* = 0.006) and ghrelin (r(54) = −0.382, *p _adjusted_* = 0.008) as well as BMI-SDS (r(63) = 0.356, *p _adjusted_* = 0.006), BDI-II (r(63) = −0.363, *p _adjusted_* = 0.009), and EDI-2 scores (r(60) = −0.284, *p _adjusted_* = 0.030). However, within-group analyses (AN, HC) showed no significant associations.

## 4. Discussion

To our knowledge, this is the first study to measure two sex steroid hormones in the hair of patients with AN. Our results are in line with the well-established finding of decreased plasma concentrations of progesterone in AN [6]. Measurements in hair constitute a novel, non-invasive method to study longer-term laboratory parameters in AN and could, therefore, reduce the burden in daily clinic routine since samples can also be obtained by staff with less training or even family members or the patients themselves. In addition, measurements in hair can detect endocrine changes and trends that are often difficult to assess given diurnal/cyclical fluctuations and provide data regarding basal longer-term endocrine alterations in AN. There is strong evidence that stress and severe energy deficiency lead to dysregulation of progesterone through changes in hypothalamic-pituitary secretion (as part of the HPG axis), i.e., slow pulsatile Gonadotropin-releasing hormone (GnRH) secretion and a consequent decrease in LH [7,22,40]. Some authors have suggested that sex steroid hormones play a role in the development and maintenance of eating disorder symptoms [6,41]. Animal studies have shown that progesterone has stimulating effects on food intake, notably by antagonizing the effects of estradiol [42,43,44]. In addition, peripheral sex steroids, including progesterone, can be converted by biosynthetic enzymes into the neuroactive steroid allopregnanolone (3α-5-tetrahydroprogesterone). Allopregnanolone is able to cross the blood-brain barrier and unfold its actions in the brain, where it can regulate affective symptoms by increasing gamma-aminobutyric acid (GABA) activity [45]. In AN, lower mean serum allopregnanolone was found to be associated with depression and anxiety, independent of BMI [46]. Furthermore, sex steroid hormones may modulate hippocampal neurogenesis with secondary effects on cognitive function, mood regulation, and emotional memory formation [47]. However, progesterone dysregulation and a subsequent disruption of the menstrual cycle are currently considered to be a consequence of the severe undernutrition rather than a causal factor in eating disorders [6].

The longitudinal arm of our study revealed no significant changes in hair-based progesterone and DHEA concentrations during inpatient treatment. Since DHEA did not show significant differences between groups in the cross-sectional design either, the following considerations refer to progesterone in particular. One speculative interpretation is that the hormonal imbalance of progesterone in AN improves more slowly than other undernutrition-related effects during treatment. While other parameters such as weight, leptin, and ghrelin significantly changed after 14 weeks of inpatient treatment (see Table 2), the recovery of progesterone levels appears to take longer. Although few studies found a normalization of progesterone levels in AN after treatment, our findings are in line with several studies examining resumption of menstruation in AN as a clinical hallmark of recovery [48]. Brambilla and colleagues [49] showed that in recovered individuals with a history of AN with persistent amenorrhea, progesterone concentrations were still significantly lower than expected, based on the patients’ stabilized nutritional status. This effect was seen despite a normal-range basal secretion of both gonadotropins FSH and LH. These patients had a normal BMI but had not yet reached their pre-morbid weight and continued to have decreased leptin levels. In contrast, another study [50] found that at a follow-up (after 6–8 month), five patients with a history of AN and resumption of the menstrual cycle did not differ from five patients with former AN, who were still amenorrhoeic, in either weight gain or hormonal variables, including progesterone values. Overall, there is strong evidence for a delay between weight gain and resumption of menstruation in AN [2]. In most patients with AN, menstruation seems to resume within 6 months of reaching target weight [51,52]. In our study, 5 of 29 patients experienced a recurrence of menstruation within the 90 days of inpatient treatment. Indeed, these patients had significant higher mean progesterone levels over the 90 days (*Md* = 0.55; *IQR* = 0.67) compared with patients without menstruation during inpatient treatment (*Md* = 0.14; *IQR* = 0.15; *U* = 114; *p* < 0.001), which is in line with the assumption that it may take some time (after reaching target weight) for normal hormonal balance, including normal progesterone levels, to be established.

An alternative explanation for our null finding may be alterations of hair growth in the acute phase of AN, which are likely a result of restricted food intake and hormonal adaptation to the underweight state [53,54]. Stress may also play a role [55]. Many AN patients experience some degree of hair loss (effluvium), especially in the telogen (resting) phase of hair development [56]. Consequently, the detection of recent hormonal changes in patients with AN in the acutely underweight phase may be affected. However, hair loss in AN follows a diffuse pattern with frontal predominance [57], while hair analyses typically used in psychiatric research are based on hair from the posterior vortex region. Furthermore, after initiating nutritional treatment in patients with AN (especially in the highly controlled environment of inpatient treatment), such effects are assumed to decrease quickly [58]. Future studies should combine hair-based hormonal assessments with standard methods and also quantify hair growth.

We did not find group differences in hair DHEA concentrations between AN and HC. We conducted post hoc power analyses which revealed a rather low power of 0.30 for group differences in DHEA levels, while the power in the other analyses was high (for details, see Appendix A). We therefore assume that a higher sample size would have been required for DHEA to rule out a beta error. However, previous studies on DHEA in blood samples in AN have obtained heterogeneous results, which may be in line with the null hypothesis. These results were summarized in a recently published review and meta-analysis [15]. The authors concluded that DHEA concentrations may be elevated in AN. However, only six studies evaluating DHEA levels in small samples were included in the meta-analysis, which increases the risk of bias given serum DHEA levels vary greatly depending on multiple factors. In addition, in two of the six included studies, the timing of hormone level measurement was not adjusted to the menstrual phase of the healthy control participants. This bias could possibly be avoided by measuring progesterone levels in hair samples. Further explanations for the diverging findings could be that other factors can also influence DHEA concentration; for example, Walther and colleagues [25] found an association between lower levels of hair-based DHEA and higher depressive symptomatology in a large sample of Indian women. However, the authors mention the lack of information about the use of oral contraception and other medications. In contrast, another study [59] found elevated DHEA levels in hair in association with child maltreatment, resulting in a dysregulation of the HPA-axis. This information should be considered in the future as maltreatment is associated with more AN symptoms [60].

The interpretation of this study has to be considered given the following limitations: First, AN-related alterations in hair growth might have affected our results, as discussed above. Second, our hair-based samples revealed a high number of progesterone and DHEA concentrations below LOD. Hormone levels are more likely to be non-detectable when truly low levels of that hormone are present [37]. In accordance with current standards in the field, which advise against removing these non-detectable/left-censored concentrations from analyses because doing so can lead to systematic bias, we did not exclude nondetectable progesterone and DHEA values but performed additional analyses that distinguished between values above and below the threshold [25]. Third, since the purpose of our study was to examine the changes in hair hormones during inpatient treatment, the treatment period may have been too short to detect significant improvements in hormone levels. In addition, our study lacks information on patients’ body fat percentage and the extent to which a sufficient level of body fat was achieved after 14 weeks of inpatient treatment, which remains an important predictor of menstrual resumption [21]. Fourth, we acknowledge that the current study follows a narrow-spectrum, hypothesis-driven approach by only examining two specific sex steroid hormones in hair and multiple interactions effects between hormones along the HPA/HPG axes (in particular cortisol) have been described [61]. Finally, we acknowledge that the sample size of our study is relatively small. Future (multi-center) studies with larger sample sizes, more comprehensive hormone assessments and comparisons between multiple measurement methods would be useful.

## 5. Conclusions

Overall, the current study provides evidence for reduced levels of progesterone measured in hair in AN. Our findings suggest that progesterone is suppressed in severe AN and does not increase rapidly during short-term weight rehabilitation. Measurement in hair allows retrospective assessment of hormone status in recent months and avoids bias due to circadian rhythms. If our results are confirmed by other studies, a possible next step could be to investigate whether any improvements (i.e., normalization) in hormone levels can be measured several months after inpatient treatment. This would be of particular interest in patients with persistent amenorrhea and could be helpful for differential diagnostics in the absence of menstruation, even after long-term recovery from anorexia.

## Figures and Tables

**Figure 1 metabolites-13-00021-f001:**
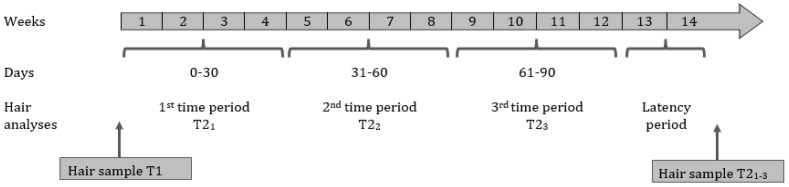
**Procedure of hair collection.** The procedure of hair collection was presented by week (1–14), by day (0–30; 31–60; 61–90), and by hair analyses period (T2_1_; T2_2_; T2_3_). T1 reflected mean hormone levels over the past 3 months before inpatient admission (pre-treatment level). T2_1–3_ reflected hormone levels during 14 weeks of inpatient treatment, including a latency period of 2 weeks to consider the time hair needs to grow above the scalp [35]. This resulted in four concentrations for hair progesterone and hair DHEA, respectively.

**Table 1 metabolites-13-00021-t001:** Sample description of the cross-sectional age matched sample.

	AN-T1 N = 33	HC N = 33	Test Statistic *T*/*X*^2^/*U*	*p*
**Demographics and clinical variables**				
Age (years)	15.0 ± 2.9	15.9 ± 2.8	−0.69	0.496
Age of onset (years)	14.4 ± 2.6	-	-	-
EDI-2 (total score) ^a^	185.4 ± 43.1	142.0 ± 34.4	4.4	<0.001 ***
BDI-II (total score)	15.4 ± 9.6	5.1 ± 5.1	5.5	<0.001 ***
SCL-90-R (global severity index)	33.9 ± 20.0	20.4 ± 19.7	2.8	<0.001 ***
**Clinical variables and plasma hormone measures**				
BMI (kg/m^2^)	14.8 ± 1.1	20.2 ± 2.1	−13.10	<0.001 ***
BMI-SDS	−2.7 ± 1.0	−0.05 ± 0.6	−12.50	<0.001 ***
Menarche (# of participants)	28	29	0.13 (*X^2^*)	0.720
Age at menarchea	12.1 ± 0.9	12.6 ± 0.9	−1.9	0.068
Menses during the last three months (# of participants) ^b^	0	29	57.0 (*X^2^*)	<0.001 ***
Leptin (ng/mL) ^c^	0.77 (1.83)	10.67 (9.11)	770.0 (*U*)	<0.001 ***
Ghrelin (ng/mL) ^c^	417.33 (265.93)	232.65 (150.90)	80.0 (*U*)	<0.001 ***
**Hair hormone measures**				
Progesterone (pg/mg)	0.07 (0.28)	0.57 (1.22)	814.0 (*U*)	<0.001 ***
DHEA (pg/mg)	0.37 (0.43)	0.40 (0.56)	646.0 (*U*)	0.193

Statistics of demographics, clinical variables, plasma hormone and hair hormone measures in the age-matched sample. AN = patients with acute anorexia nervosa; HC = healthy controls. BMI-SDS = Body mass index standard deviation score. EDI-2 = Eating Disorder Inventory 2; SCL-90-R (GSI) = Symptom Checklist-90-Revised (global severity index). All patients with AN were of the restrictive AN subtype. Group differences in demographics and clinical variables, plasma hormones and weight measures were tested parametrically using *X*^2^ or T-Tests. Differences in leptin, ghrelin and hair hormone measures were tested nonparametrically using Mann–Whitney U Tests. Mean value ± standard deviation for each variable is shown separately for each group. Leptin, ghrelin and hair hormone values are reported as the median and interquartile range (IQR) for each group. * *p* < 0.05, ** *p* < 0.01, *** *p* < 0.001 (^a^ N = 31 and 32, for AN and HC, respectively; ^b^ N = 28 and 29, for AN and HC, respectively; ^c^ N = 26 and 31, for AN and HC, respectively).

**Table 2 metabolites-13-00021-t002:** Sample description of the longitudinal AN sample.

	AN-T1 T_1_ Pre-Treatment Level	AN-T2 T2_1_ Time Period 2_1_ (Weeks 1–4 after Admission)	T2_2_ Time Period 2_2_ (Weeks 5–8 after Admission)	T2_3_ Time Period 2_3_ (Weeks 9–12 after Admission)	Test Statistic *X*^2^*_F_*/*T*	*p*
**Clinical variables and plasma hormone measures**						
Weight (kg)^a^	41.35 (7.03)	44.75 (7.3)	49.60 (7.28)	51.70 (7.28)	77.7 (X^2^_F_)	<0.001 ***
BMI (kg/m^2^) ^a^	15.16 (1.52)	16.30 (1.75)	17.69 (2.10)	18.96 (1.97)	77.7 (X^2^_F_)	<0.001 ***
BMI-SDS ^a^	−2.58 (1.19)	−1.74 (1.28)	−1.11 (1.11)	−0.54 (.64)	77.7 (X^2^_F_)	<0.001 ***
Menses during the last three months (# of participants) ^b^	0	-	-	5	-	-
Leptin (ng/mL) ^c^	1.04 (2.00)	-	-	10.37 (11.86)	9.0 (T)	<0.001 ***
Ghrelin (ng/mL) ^c^	441.50 (305.28)	-	-	213.70 (150.06)	210.0 (T)	<0.001 ***
**Hair hormone measures**						
Progesterone ^b^	0.07 (0.25)	0.07 (0.37)	0.07 (0.22)	0.07 (0.30)	1.5 (X^2^_F_)	0.682
DHEA ^b^	0.37 (0.25)	0.43 (0.25)	0.43 (0.25)	0.38 (0.29)	2.3 (X^2^_F_)	0.509

Statistics of clinical variables, plasma hormone and hair hormone measures during the course of inpatient treatment. AN = patients with acute anorexia nervosa; BMI = body mass index; BMI-SDS = BMI standard deviation score. Medians and interquartile range (IQR) for each variable are shown separately for each time period. Weight, BMI and BMI-SDS were collected 30, 60 and 90 days after admission. Time difference of blood collection between T1 and T2 for leptin and ghrelin ranged from 74 to 154 days. Differences were tested nonparametrically using the Wilcoxon signed rank test or Friedman’s test. Hair hormone concentrations of progesterone and DHEA at T1 represent the pre-treatment level (last three months). *** *p* < 0.001 (^a^ N = 26, ^b^ N = 29, ^c^ N = 20).

## Data Availability

Mass spectrometry protocols (LC-MS/MS) are published by Gao and colleagues (2013): Gao, W.; Stalder, T.; Foley, P.; Rauh, M.; Deng, H.; Kirschbaum, C. Quantitative Analysis of Steroid Hormones in Human Hair Using a Column-Switching LC–APCI–MS/MS Assay. *J. Chromatogr. B* 2013, *928*, 1–8, doi:10.1016/j.jchromb.2013.03.008. The data presented in this study are available on request from the corresponding author. The data are not publicly available due to ethics comittee requirements.

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
