# Peer review of "Hair-Based Assessment of Sex Steroid Hormones in Patients with Anorexia Nervosa"

_metabolites, 2022, doi:10.3390/metabo13010021_

Round 1
Reviewer 1 Report
The authors have presented an interesting and novel approach for evaluating sex hormone levels in patients with anorexia nervosa. This type of longitudinal approach in hair measurement has proven quite useful in other applications such as adherence to drug treatment protocols and detection of drugs of abuse.
In the case of AN, however, hair may not grow normally, or at all. Thus any assumptions based on time using typical hair growth are likely to be inaccurate. While this problem is mentioned in the final paragraph of the discussion, it is not adequately addressed. This could very well be the reason that no change in progesterone was detected over the course of the in patient treatment. This should be discussed in more detail in the manuscript. Could this be mitigated by measuring hair growth in patients? If patients' hair is growing, it might be uneven as their BMI and nutrition levels improve. Thus, equal segmentation may not be appropriate.
Also, no mention at all of the LC-MS method is given in the main manuscript and only a reference from where LC-MS methods were taken is given in the supplemental information. The reader should not have to be twice redirected to find out how the study was performed. At least some detail should be given in the manuscript.
Reviewer 2 Report
This is a very well-written manuscript, and the authors have conducted a well-structured study. I agree with the authors that hormonal assets in AN are still understudied. I have only a few comments that I hope might help the authors, then I think the manuscript might be accepted.
- Have you evaluated the sample size needed for this study?
- Have you evaluated other hormonal levels, for example, cortisol? It has a role in the regulation of menses, and in ED patients, it has been several times shown to be alternated (see https://doi.org/10.1002/eat.22375 or https://doi.org/10.1002/erv.2896). For example, it might have a role in longitudinal non-improvement.
Round 2
Reviewer 1 Report
The manuscript looks good. I have no further comments.
Author Response
We are grateful that the Reviewer was satisfied with our answers and the current manuscript. We hope that it can now be accepted for publication in the Journal.